# PLOD2 Is a Prognostic Marker in Glioblastoma That Modulates the Immune Microenvironment and Tumor Progression

**DOI:** 10.3390/ijms23116037

**Published:** 2022-05-27

**Authors:** Nina Kreße, Hannah Schröder, Klaus-Peter Stein, Ludwig Wilkens, Christian Mawrin, Ibrahim Erol Sandalcioglu, Claudia Alexandra Dumitru

**Affiliations:** 1Department of Neurosurgery, Otto-von-Guericke University, 39120 Magdeburg, Germany; nina.kresse@st.ovgu.de (N.K.); hannah.schroeder@st.ovgu.de (H.S.); klaus-peter.stein@med.ovgu.de (K.-P.S.); erol.sandalcioglu@med.ovgu.de (I.E.S.); 2Department of Pathology, Nordstadt Hospital Hannover, 30167 Hannover, Germany; ludwig.wilkens@krh.eu; 3Department of Neuropathology, Otto-von-Guericke University, 39120 Magdeburg, Germany; christian.mawrin@med.ovgu.de

**Keywords:** glioblastoma, PLOD2, prognostic biomarkers, tumor progression, neutrophils

## Abstract

This study aimed to investigate the role of Procollagen-Lysine, 2-Oxoglutarate 5-Dioxygenase 2 (PLOD2) in glioblastoma (GBM) pathophysiology. To this end, PLOD2 protein expression was assessed by immunohistochemistry in two independent cohorts of patients with primary GBM (*n*_1_ = 204 and *n*_2_ = 203, respectively). Association with the outcome was tested by Kaplan–Meier, log-rank and multivariate Cox regression analysis in patients with confirmed IDH wild-type status. The biological effects and downstream mechanisms of PLOD2 were assessed in stable PLOD2 knock-down GBM cell lines. High levels of PLOD2 significantly associated with (*p*_1_ = 0.020; *p*_2_
*<* 0.001; log-rank) and predicted (cohort 1: HR = 1.401, CI [95%] = 1.009–1.946, *p*_1_ = 0.044; cohort 2: HR = 1.493; CI [95%] = 1.042–2.140, *p*_2_ = 0.029; Cox regression) the poor overall survival of GBM patients. PLOD2 knock-down inhibited tumor proliferation, invasion and anchorage-independent growth. MT1-MMP, CD44, CD99, Catenin D1 and MMP2 were downstream of PLOD2 in GBM cells. GBM cells produced soluble factors via PLOD2, which subsequently induced neutrophils to acquire a pro-tumor phenotype characterized by prolonged survival and the release of MMP9. Importantly, GBM patients with synchronous high levels of PLOD2 and neutrophil infiltration had significantly worse overall survival (*p* < 0.001; log-rank) compared to the other groups of GBM patients. These findings indicate that PLOD2 promotes GBM progression and might be a useful therapeutic target in this type of cancer.

## 1. Introduction

GBM is the most common and fatal malignant primary brain tumor in adults [1]. Despite the extensive standard of care therapy including maximal safe surgical resection followed by radiation and chemotherapy, the relative 5-year survival of GBM patients is less than 7% with a median survival of 14 months [1,2,3]. The diffuse infiltration in the surrounding brain parenchyma makes complete surgical resection impossible and is the main reason for recurrence and therapy resistance of GBM (reviewed in [4]). Therefore, the focus of research regarding therapeutic approaches for GBM has shifted towards immunotherapy and individualized therapy. In particular, numerous vaccine approaches, oncolytic viruses and immune-checkpoint inhibitors are in preclinical and clinical trials [3,5]. Furthermore, specific targeted therapies gained importance through a better understanding of the underlying molecular heterogeneity of GBM. Despite these multimodal approaches, GBM remains an incurable disease at present. Thus, there is still an urgent need to identify novel cellular and molecular mechanisms that control the progression of GBM and could serve as therapeutic targets in this type of cancer.

Accumulating evidence indicates that the interplay between tumor cells and the tumor microenvironment (TME) plays a crucial role in tumor migration, invasion and progression [6,7]. The TME is largely determined by the extracellular matrix (ECM) with collagen as the most abundant protein [8]. During tumor progression, increased collagen crosslinking promotes stiffening of the extracellular matrix, thus enhancing invasion and metastasis [9,10]. The main enzyme mediating stabilized collagen crosslinks is Procollagen-Lysine,2-Oxoglutarat 5-Dioxygenase 2 (PLOD2) [11]. This membrane-bound homodimeric enzyme hydroxylates lysine residues in the telopeptides of procollagens and thus, plays a crucial role in the post-translational modification of collagen biosynthesis [11]. The resulting hydroxyl groups are essential for the formation of stable crosslinks by lysyl oxidases [12].

PLOD2 is upregulated in various cancers and is associated with poor outcomes in bladder cancer [13], hepatocellular carcinoma [14,15] and breast cancer [16] (and reviewed in [17]). Exploratory studies on a small cohort of 28 GBM patients indicated that PLOD2 was also associated with poor survival in this type of cancer [18]. Furthermore, Xu et al. found that high gene expression of PLOD2 was significantly associated with poor overall- and progression-free survival in glioblastoma patients [19]. The same study also showed that increasing PLOD2 protein levels were associated with increasing tumor grade in glioma [19]. At the molecular level, PLOD2 induces epithelial-mesenchymal transition [20] and activates the PI3K-Akt [20], JAK-STAT [21] and FAK [19] signaling pathways. Although the exact mechanisms are still largely unclear, PLOD2 can be expected to affect key signaling pathways in tumor cells, thereby modulating tumor progression.

The TME also contains a variety of infiltrating and resident immune cells that interact with the tumor cells and modulate their biology and functions. Recent studies showed that the GBM microenvironment hosts a large number of tumor-infiltrating neutrophils, which are actively recruited by GBM cells through the expression of IL-8 and IL-1b [22,23] (and reviewed in [24]). Importantly, the presence of infiltrating neutrophils in GBM was significantly associated with a poor outcome in these patients (reviewed in [24]). These findings suggest that neutrophils are substantially involved in the progression of GBM. As an important modulator of the extracellular matrix and TME, PLOD2 may also activate the tumor-infiltrating neutrophils.

The role of PLOD2 in the pathophysiology of GBM still requires extensive characterization. This study aimed to determine (1) the association between PLOD2 expression and the clinical outcome of IDH wild-type GBM patients; (2) the involvement of PLOD2 in the modulation of GBM tumor cell functions and (3) the effect of PLOD2 on the biology and function of neutrophils.

## 2. Results

### 2.1. PLOD2 Associates with and Predicts Poor Overall Survival of GBM Patients

Previous studies by Xu et al. using the TCGA database showed that high gene expression of PLOD2 is significantly associated with a poor outcome in GBM patients [19]. Here, we investigated whether the protein levels of PLOD2 in tumor tissue are associated with overall survival (OS) or progression-free survival (PFS) of GBM patients with confirmed IDH wild-type (IDH WT) status. To this end, the levels of PLOD2 were assessed by immunohistochemistry (see Material and Methods section) in two independent patient cohorts. PLOD2 expression was subsequently dichotomized into “low“ and “high“ based on the median-split method. The survival curves were plotted according to the Kaplan–Meier method and the statistical significance was assessed with the log-rank test. In the Hannover cohort, GBM patients with high tumor levels of PLOD2 (PLOD2^high^) had a significantly shorter OS compared to patients with low levels of PLOD2 (PLOD2^low^) (*p* = 0.020; log-rank) (Figure 1A). These findings were confirmed in the Magdeburg cohort of GBM patients (*p* < 0.001; log-rank) (Figure 1B). In both cohorts, PLOD2^high^ patients had a shorter PFS compared to their PLOD2^low^ counterparts, but statistical significance was only reached in the Magdeburg cohort (*p* = 0.001; log-rank) (Figure 1C, D).

We further analyzed the OS and PFS of IDH WT GBM patients using Cox proportional-hazard models adjusted for factors known to influence the patients’ outcome, such as age [25], Karnofsky Performance Scale (KPS) [26], extent of surgical resection [27], therapy [28] and MGMT methylation status [29]. Initial analysis of the time-dependent covariate (T_COV_) for PLOD2 showed that the proportional hazard assumption of these models had been satisfied: Hannover cohort_OS: *p* = 0.272; Magdeburg cohort_OS: *p* = 0.448; Hannover cohort_PFS: *p* = 0.440; Magdeburg cohort_PFS: *p* = 0.402. In both cohorts, high expression of PLOD2 predicted poor OS in GBM patients (Hannover cohort: HR = 1.401, CI [95%] = 1.009–1.946, *p* = 0.044; Magdeburg cohort: HR = 1.493, CI [95%] = 1.042–2.140, *p* = 0.029) (Figure 2A). For PFS, PLOD2^high^ patients had an increased hazard ratio compared to PLOD2^low^ patients but statistical significance was only reached in the Magdeburg cohort (HR = 1.645, CI [95%] = 1.040–2.603, *p* = 0.033) (Figure 2B). These data indicate that PLOD2 could serve as an independent prognostic biomarker, at least for the overall survival of GBM patients.

### 2.2. PLOD2 Promotes the Invasion, Proliferation and Anchorage-Independent Growth of GBM Cells

Using U87 and U251 GBM cells, recent studies showed that PLOD2 enhanced the aggressiveness of GBM cells by promoting tumor invasion [19,20]. Here we investigated the effect of PLOD2 on the biology and functions of GBM using the H4 GBM cell line. To this end, the cells were stably transfected with a sh-RNA plasmid to downregulate the levels of PLOD2 (sh-PLOD2) or with a control plasmid (sh-control) (see Material and Methods section). All functional assays were performed in the absence of the selection antibiotic puromycin. Control western blot analysis confirmed that PLOD2 knock-downs remained stable until at least day 10, which was the last time point of the longest assay (Appendix A).

Tumor invasion was assessed by the degree of “gap” closure (red line) in a 3D collagen matrix, using the Oris^TM^ system (Figure 3A). The results showed that PLOD2 knock-down significantly reduced the invasiveness of H4 GBM cells (Figure 3B). We additionally determined the activity of matrix metalloproteases (MMP2 and MMP9), since MMPs are critical for tumor invasion in many types of cancer, including GBM [30]. To this end, sh-control and sh-PLOD2 GBM cells were incubated in a culture medium and the supernatants were collected 48 h later. A culture medium without cells was used as a control. We found that H4 GBM cells released MMP2 but only negligible levels of MMP9 (Figure 3C). Importantly, PLOD2 knock-down cells released significantly lower levels of MMP2 compared to their control-transfected counterparts (Figure 3D). These findings indicate that PLOD2 enhances the invasiveness of GBM cells, possibly via MMP2.

In further studies, we investigated the role of PLOD2 in GBM proliferation by assessing the metabolic activity of the transfected H4 cells using the MTT assay. To this end, the concentration of metabolized MTT was measured at different time points in one setup with 2000 cells and another with 4000 cells. The results showed that PLOD2 knock-down decreased the metabolic activity of H4 cells in both setups (Figure 4A,B). However, statistical significance was only reached for certain time points. We additionally determined the anchorage-independent growth of transfected GBM cells by allowing the cells to form colonies in low-gelling agarose for 10 days (Figure 4C). We found that PLOD2 knock-down cells formed significantly fewer colonies than their sh-control counterparts (Figure 4D). Taken together, these data indicate that PLOD2 promotes the invasion, proliferation and anchorage-independent growth of GBM cells.

### 2.3. PLOD2 Modulates the Expression of Catenin D1, CD44, CD99 and MT1-MMP in GBM Cells

As shown above, PLOD2 modulates the biological functions of GBM cells. To obtain further insight into the molecular mechanisms downstream of PLOD2 in GBM cells, we assessed by western blot the protein expression of several markers associated with tumor proliferation and invasion, such as Catenin D1, CD44, CD99, CDK6, EGFR, HIF1-beta, Integrin beta-1, MT1-MMP and PRAS40. We found that the levels of Catenin D1, CD44, CD99 and MT1-MMP were significantly lower in PLOD2 knock-down cells compared to their control-transfected counterparts (Figure 5A–E).

### 2.4. GBM-Associated PLOD2 Induces Neutrophil Granulocytes to Aquire a Pro-Tumor Phenotype

Accumulating evidence indicates that the GBM microenvironment contains significant numbers of neutrophils and that high neutrophil infiltration is associated with poor outcomes in GBM patients (reviewed in [24]). Furthermore, very recent studies found a correlation between high tumor levels of PLOD2 and high neutrophil infiltration in cervical [31] and hepatocellular carcinoma [14] tissues. Based on these findings, we hypothesized that GBM cells modulate the biology and functions of neutrophils via PLOD2. To test this hypothesis, we produced conditioned supernatants (SN) from sh-control and sh-PLOD2 GBM cells. Subsequently, we stimulated peripheral blood neutrophils with these supernatants and determined neutrophil survival as well as the release of MMP9—both indicators of a pro-tumor neutrophil phenotype (Figure 6A).

The results showed that the sh-control SN from H4 cells prolonged the survival of neutrophils at 24 h post-stimulation (Figure 6B). This effect was significantly lower upon stimulation with sh-PLOD2 H4 SN (Figure 6B). To confirm these findings, we additionally stimulated neutrophils with SN from a second GBM cell line (U251). Similar to the H4 SN, the sh-control U251 SN prolonged neutrophil survival while sh-PLOD2 U251 SN had a significantly weaker effect (Figure 6B). To test whether GBM cells induce neutrophils to release MMP9, we stimulated neutrophils with sh-control or sh-PLOD2 SN for 1 h and determined MMP9 release by gelatin zymography. We found that the sh-control SN from both H4 and U251 cells induced neutrophils to release MMP9 (Figure 6C,D). The release of MMP9 by neutrophils was significantly lower upon stimulation with sh-PLOD2 SN (Figure 6C,D). GBM SN without neutrophils had only negligible levels of MMP9 (data not shown). To exclude potential clonal effects, we repeated this set of studies with SN derived from different sh-PLOD2 clones—for both H4 and U251 cells—and obtained similar results (Appendix A). Taken together these findings indicate that GBM cells release soluble factors via PLOD2, which stimulate neutrophils to acquire a tumor-promoting phenotype.

To test the clinical relevance of these findings, we stained GBM tissues against the neutrophilic marker CD66b. The patients were subsequently divided into four groups according to the combined expression of CD66b and PLOD2: CD66b^low^/PLOD2^low^, CD66b^low^/PLOD2^high^, CD66b^high^/PLOD2^low^ and CD66b^high^/PLOD2^high^. Kaplan–Meier analysis revealed that CD66b^high^/PLOD2^high^ patients had the shortest overall survival of all GBM patients (Figure 6E). These results were confirmed in a multivariate Cox regression model adjusted for age, KPS, therapy, resection efficiency and MGMT status where CD66b^high^/PLOD2^high^ patients had a significantly increased hazard ratio compared to the other groups of GBM patients (HR = 1.703, CI [95%] = 1.067–2.720, *p* = 0.026) (Figure 6F).

## 3. Discussion

An increased effort has been made to identify the cellular/molecular factors that modulate the pathophysiology of GBM and that could provide information regarding diagnosis, prognosis and therapy in this type of cancer. PLOD2 is a promising biomarker and a target for cancer therapy, but its exact role in GBM still requires characterization. Several studies found an association between PLOD2 overexpression and poor outcome in multiple types of cancer, such as sarcoma [32], breast cancer [16], hepatocellular carcinoma [14] and bladder cancer [13]. Previous studies on a small (*n* = 28) cohort of GBM patients suggested that PLOD2 may serve as a biomarker in this type of cancer [18]. Song et al. found an association between high PLOD2 expression and poor outcomes in glioma patients. However, their study did not distinguish between high grade and low grade gliomas [20]. In a comprehensive study on glioma and GBM patients, Xu et al. demonstrated that increasing PLOD2 protein levels are associated with increasing tumor grade. Furthermore, the gene expression of PLOD2 was significantly higher in GBM than in healthy tissues and correlated with overall and progression-free survival [19]. Using two independent cohorts of IDH WT GBM patients, our study shows that high protein levels of PLOD2 (PLOD2^high^) are significantly associated with and predicted for poor overall survival of these patients. Together with the study by Xu et al., these findings indicate that PLOD2 could be a robust biomarker for the survival of GBM patients.

We next sought to characterize the biological functions of PLOD2 in GBM cells. We found that PLOD2 promoted the invasiveness of H4 GBM cells. These data are in line with previous studies showing that PLOD2 modulates the migration and invasion of glioma cells. Specifically, Song et al. showed that PLOD2 knock-down suppressed the migration and invasion of U87 and U251 GBM cells, while Xu et al. showed in the same cell lines that the depletion of PLOD2 decreased invasion in vitro and in vivo, possibly by remodeling the stiffness of the ECM and decreasing the focal adhesion plaques [19,20]. We additionally found that PLOD2 promoted the release of ECM-degrading MMP2—a mechanism associated with enhanced invasiveness and worse outcomes in different types of cancer, including glioma [33,34,35]. Furthermore, we demonstrate that PLOD2 promotes the metabolic activity and the anchorage-independent growth of GBM cells. The effect of PLOD2 on the anchorage-independent tumor growth was especially striking since PLOD2 knock-down led to almost a complete inhibition of colony formation in GBM cells. Together, these findings are of particular importance for the pathophysiology of GBM, since high tumor invasiveness into the adjacent brain tissue and rapid growth are the main reasons why these tumors remain incurable at present. It should be mentioned at this point, that regulation of the ECM and tumor invasion is extremely complex. PLOD2 alone seems to play multiple roles in this process, since it can both degrade the basement membrane via MMP2 release (our own data), as well as induce collagen crosslinking/stiffening, thereby creating a “highway” for local invasion and activating different signaling pathways by mechanotransduction [6,17,19]. Furthermore, as elegant studies by Georgescu et al. recently showed, there are many other factors modulating the ECM program in the microenvironment of GBM [36]. Thus, the mechanisms involved in GBM invasion and the exact role of PLOD2 in this process still require further characterization.

Previous studies found an association between PLOD2 and epithelial-mesenchymal transition (EMT), hypoxia-induced activation of PI3K-Akt signaling, as well as FAK phosphorylation in GBM cells [19,20]. Here, we found that PLOD2 knock-down cells had decreased levels of MT1-MMP, which is known to be a key regulator of cell migration and invasion in GBM [37,38,39]. Furthermore, MT1-MMP is an important activator of MMP2 leading to an enhanced invasion and progression in different tumor types including GBM [37,40,41,42]. These findings support our data on the PLOD2-mediated release of MMP2 in GBM cells (see above). We additionally found that PLOD2 knock-down decreased the levels of CD44 indicating that PLOD2 regulates CD44 expression. CD44 is known to promote tumor formation through interactions with the tumor microenvironment and is involved in various cellular processes including invasion, proliferation and apoptosis in many types of cancer including GBM (reviewed in [43]). Increased CD44 expression was associated with worse survival in GBM [44]. By which mechanisms PLOD2 affects CD44 expression in GBM remains unclear. However, previous studies in laryngeal carcinoma suggested that PLOD2 enhanced CD44 expression via activation of the Wnt-signaling pathway [45], which may be a possible explanation for GBM as well. Our study further found that PLOD2 knock-down decreased the levels of CD99 in GBM cells. CD99 is known to alter the structure of the cytoskeleton, thus facilitating cell migration [46,47]. Indeed, studies on the GBM cell line U87 showed that CD99 overexpression increased the migration and invasion of these cells [46,47]. The exact mechanisms of CD99 regulation by PLOD2 are, however, currently unknown and remain to be characterized in future studies. Finally, our data showed that Catenin D1 levels were lower upon PLOD2 knock-down. Catenin D1 has been previously linked to oncogenic signaling pathways important for anchorage-independent cell growth [48]. This supports our findings that PLOD2 knock-down cells were almost completely unable to form colonies in soft agar clonogenic assays.

Increasing evidence suggests that the interplay between tumor cells and the immune system is a key modulator of tumor biology and determines cancer pathogenesis and progression. Recent studies found a significant correlation between PLOD2 and infiltrating immune cells including neutrophils in cervical, hepatocellular and lung cancer [14,31,49]. These findings suggest that PLOD2 is an important modulator of the tumor immune microenvironment. The GBM microenvironment contains high numbers of infiltrating neutrophil granulocytes [23], which were found to associate with a poor outcome in GBM patients [22] (and reviewed in [24]). Neutrophils release various factors in the microenvironment, which can promote tumor progression. For instance, neutrophils have strong pro-angiogenetic activity via the release of MMP9 and vascular endothelial growth factor (VEGF). Additionally, neutrophils promote tumor motility, migration and invasion via the release of neutrophil elastase, cathepsin G, proteinase 3, MMP8 and MMP9. Under physiological conditions, neutrophils rapidly undergo apoptosis. However, their lifespan can be prolonged by tumor-derived factors resulting in enhanced local inflammation and, ultimately, tumor progression [50]. Our study shows that PLOD2 controls the production of (currently unidentified) soluble factors by GBM cells, which subsequently enhance neutrophil survival and the release of MMP9. Moreover, patients with synchronous high expression of PLOD2 and the neutrophilic marker CD66b had a significantly shorter overall survival compared to the other groups of GBM patients. These data suggest that PLOD2 modulates the immune microenvironment of GBM leading to the progression of this cancer.

All of the above supports the fact that PLOD2 promotes GBM progression and, thus, might serve as a potential therapeutic target. Several pharmacological inhibitors of PLOD2 are available at present. In particular, Minoxidil was found to reduce sarcoma migration and metastasis in vitro and in vivo by inhibiting PLOD family members [32]. Similarly, inhibition of PLOD2 by Minoxidil reduced tumor migration in lung carcinoma and might prevent metastasis in this type of cancer [51]. Interestingly, several studies additionally found that Minoxidil enhanced drug delivery, including that of Temozolomide, by permeabilizing the blood-tumor barrier (BTB) in GBM [52,53]. It would be, therefore, tempting to speculate that GBM patients with high expression of PLOD2 might benefit from individualized therapies with PLOD2 inhibitors, such as Minoxidil.

In summary, our study identifies PLOD2 as an independent prognostic biomarker in GBM. Furthermore, we demonstrate that PLOD2 mediates important biological functions of GBM cells, such as proliferation, invasion and anchorage-independent growth. We additionally show that PLOD2 regulates the expression of MMP2, MT1-MMP, CD44, CD99 and Catenin D1 in GBM cells. Importantly, we link PLOD2 with the immune modulation of neutrophils in the microenvironment of GBM. The main results of our study are summarized in Figure 7. These findings contribute to a better understanding of GBM pathophysiology and may ultimately foster the development of novel therapeutic strategies against this type of cancer.

## 4. Materials and Methods

### 4.1. Study Subjects

In this study, we retrospectively analyzed tissues from two independent cohorts of adult patients with histopathologically confirmed, newly diagnosed GBM. The tumors were clinically classified as primary GBM, as no lower grade glioma had been documented in the patient’s medical history. The patients in the Hannover cohort were treated at the Department of Neurosurgery, Nordstadt Hospital Hannover between 2004–2014 and had a median age of 66 years. The patients in the Magdeburg cohort were treated at the Department of Neurosurgery, University Hospital Magdeburg between 2005–2018 and had a median age of 64 years. All studies were carried out in accordance with the Declaration of Helsinki of 1975, revised in 2013 and approved by the ethics committees of the Medical School Hannover (Study Nr. 6864, 2015) and Otto-von-Guericke University Magdeburg (Study Nr. 146, 2019), respectively. The ethics committees additionally provided a waiver for the need for informed consent. The clinical characteristics of the patients including sex, post-operative Karnofsky Performance Scale (KPS), therapy, extent of surgical resection, MGMT methylation status and IDH mutation status are summarized in Appendix A. The survival analysis was performed only on patients with confirmed IDH wild-type (WT) status. The clinical characteristics of the IDH WT patients are separately summarized in Appendix A.

### 4.2. Tissue Microarrays (TMA): Immunhistochemistry and Scoring

TMA blocks were built using the Arraymold kit E (Riverton, UT, USA) as previously described [54,55] and cut into 2 µm sections. The sections were incubated with 667 ng/mL PLOD2-specific polyclonal antibodies (Proteintech Europe, Manchester, UK) or 500 ng/mL anti-human CD66b antibodies (BioLegend, San Diego, CA, USA) at 4 °C overnight. Secondary and colorimetric reactions were performed using the UltraVision^TM^ Detection System according to the manufacturer’s instructions (Thermo Scientific, Freemont, CA, USA). Nuclei were counterstained with Haematoxylin (Carl Roth, Karlsruhe, Germany) and the sections were covered with Mountex^®^ embedding medium (Medite, Burgdorf, Germany). All stained TMAs were digitalized with an Aperio VERSA 8 high-resolution whole-slide scanner and the digital images were viewed with the Aperio ImageScope software (Leica Biosystems, Nussloch, Germany). Authors C.A.D., H.S. and N.K. independently performed blinded histological analysis.

PLOD2 exhibited mainly a cytoplasmic subcellular localization. The expression intensity of the marker was categorized as “weak”, “medium” or “strong” and assigned 1, 2, or 3 points, respectively (Appendix A). As a number of samples exhibited heterogeneous staining, the expression was subsequently graded using the H-Score according to the formula:(1 × X) + (2 × Y) + (3 × Z), where X + Y + Z = 100% of the total tumor area(1)

CD66b (neutrophilic marker) was assessed by counting the number of positive cells at 20× magnification in at least two different fields per TMA spot. Samples with an average of ≤5 cells/field were considered as “CD66b^low^” and samples with >5 cells/field as “CD66b^high^” (Appendix A).

### 4.3. Cell Lines and Stable Transfection

Both H4 and U251 cell lines were a kind gift from Prof A. Temme (University Hospital Dresden) but are also commercially available. The main characteristics of these cells are shown in Appendix A. The cells were cultured in Dulbecco’s Modified Eagle Medium (Gibco^®^ DMEM; Thermo Fisher Scientific, Dreieich, Germany) supplemented with 10% fetal calf serum (FCS; Pan Biotech, Aidenbach, Germany), and 1% Penicillin-Streptomycin (Gibco^®^ DMEM; Thermo Fisher Scientific). The cells were transfected with the OmicsLink^TM^shRNA clone HSH013271-nH1-b against PLOD2 or with CSHCTR001-nH1 as a control (both from GeneCopoeia, Rockville, MD, USA) using Panfect A-plus transfection reagent (Pan Biotech) according to the manufacturer’s protocol. Transfected cells were selected with 1 µg/mL Puromycin (InvivoGen, Toulouse, France) and then maintained in cell culture medium containing 0.3 µg/mL Puromycin. The efficiency of PLOD2 knock-down (sh-PLOD2) compared to control transfection (sh-control) was assessed both at protein and mRNA levels (Appendix A). Based on these results, clone 6 (C6) for H4 and clone 3 (C3) for U251 cells were used in all subsequent experiments. To exclude potential clonal effects, selected experiments were performed using clone 4 (C4) for H4 and clone 1 (C1) for U251 cells (Appendix A). In all figures depicting in vitro studies, “+” indicates the sample where the respective cells (either sh-PLOD2 or sh-control) were used.

### 4.4. SDS-PAGE and Western Blot

GBM cells were lysed with a commercially available buffer containing Triton X-100 and protease/phosphatase inhibitors (both from Cell Signaling Technology, Frankfurt am Main, Germany). Cell debris was removed by centrifugation and the lysates were incubated with an SDS-Loading buffer containing 4% glycerin, 0.8% SDS, 1.6% beta-mercaptoethanol and 0.04% bromophenol blue (all from Carl Roth). Samples were separated by SDS-PAGE followed by transfer to Immobilon-P (Merck Millipore) or Roti^®^-Fluoro (Carl Roth) PVDF membranes. The membranes were incubated with the following primary antibodies: anti-Catenin D1, anti-CD44, anti-CD99 (from Cell Signaling Technology), anti-MT1-MMP and anti-PLOD2 (from Proteintech) overnight at 4 °C. Secondary reactions were performed for 1h at room temperature using HRP-, AlexaFluor^®^488- or AlexaFluor^®^647-coupled antibodies (all from Cell Signaling Technology). All antibodies were diluted as recommended by the respective manufacturer using the SignalBoost™ Immunoreaction Enhancer Kit (Merck Millipore). Signal detection was performed on a ChemoStar imaging system (Intas Science Imaging, Göttingen, Germany).

### 4.5. Gene Expression Analysis

The mRNA from sh-PLOD2 and sh-control GBM cells was isolated with the InnuPREP RNA Mini Kit 2.0 (Analytik Jena AG, Jena, Germany) according to the manufacturer’s instructions. Subsequently, reverse transcription was performed with the LunaScript RT SuperMix Kit (New England Biolabs, Frankfurt am Main, Germany). The samples were incubated with primers against PLOD2 or GAPDH in the presence of Luna Universal qPCR Mix (New England Biolabs). The following primers were used:

PLOD2 forward 5′- CATGGACACAGGATAATGGCTG-3′

PLOD2 reverse 5′-AGGGGTTGGTTGCTCAATAAAA-3′

GAPDH forward 5′-AGGGCTGCTTTTAACTCTGGT-3′

GAPDH reverse 5′-CCCCACTTGATTTTGGAGGGA-3′

### 4.6. MTT Assay

GBM cells were seeded at a density of 2000 cells/well and 4000 cells/well in 96-well plates. At the indicated time-points, fresh medium containing 10% MTT (3-(4, 5-dimethylthiazol-2-yl)-2, 5-diphenyltetrazolium bromide) (Carl Roth) was added and the samples were incubated for 4 h at 37 °C to allow for the formation of formazan crystals. After lysis with a solution containing isopropanol and hydrochloric acid (both from Carl Roth), colorimetric detection was performed at OD_540_-OD_690_ on a TECAN plate reader (Tecan, Männedorf, Switzerland).

### 4.7. Soft Agar Clonogenic Assay

Ninety-six-well plates were coated with 1% high-gelling agarose (Carl Roth). GBM cells (1000 cells/well) were mixed with low-gelling agarose (Carl Roth) at a final concentration of 0.3% and were added on top of the first layer. The low-gelling agarose was allowed to solidify for 1 h at 4 °C and culture medium was added to each well. The samples were incubated at 37 °C for 10 days with medium change every 3–4 days. The samples were subsequently stained with a solution containing 0.05% Crystal Violet (Carl Roth). Colonies with a diameter of at least 50 µm were counted using a BZ-X810 microscope (Keyence, Neu-Isenburg, Germany).

### 4.8. Invasion Assay

The invasion of GBM cells was assessed with the ORIS^TM^ cell invasion system (Platypus Technologies LLC, Madison, WI, USA) according to the manufacturer’s instructions. The GBM cells were allowed to invade for 72 h in a matrix containing 1 mg/mL collagen I. The degree of “gap-closure” was quantified with the ImageJ 1.48v software.

### 4.9. Gelatin Zymography

The release of matrix metalloproteases (MMPs) by GBM cells was analyzed by gelatin zymography, as described previously [56]. Briefly, 10^5^ cells/mL were incubated at 37 °C in DMEM medium, supplemented as above. As serum-supplemented cell culture medium also contains MMPs, medium without cells was used as control. The supernatants were collected at 48 h and mixed with Zymogram sample buffer at a final concentration of 80 mM Tris pH 6.8, 1% SDS, 4% glycerol and 0.006% bromophenol blue. Proteins were separated by SDS-PAGE containing 0.2% gelatin 180 Bloom and then renatured in 2.5% Triton-X-100 for 1 h at room temperature. The enzymatic reaction was performed overnight at 37 °C in a buffer containing 50 mM Tris pH 7.5, 200 mM NaCl, 5 mM CaCl_2_ and 1% Triton-X-100. The gels were stained with a solution containing 0.5% Coomassie blue, 30% methanol and 10% acetic acid for 1 h at room temperature. Finally, the gels were de-stained with 30% methanol and 10% acetic acid until the digested bands became visible. All chemicals were from Carl Roth (Karlsruhe, Germany). The gelatinolytic bands were quantified with ImageJ 1.48v software. The release of MMPs by neutrophils was assessed as above, except for using a different cell number (10^6^ cells/mL) and duration of stimulation (1 h).

### 4.10. Isolation of Neutrophils from Peripheral Blood

Diluted blood (1:1, *v*/*v* in phosphate buffered saline (PBS)) was subjected to density gradient centrifugation using Pancoll (Pan Biotech). The mononuclear cell fraction was discarded, and the neutrophil fraction was collected in a fresh test tube. Erythrocytes were removed by sedimentation with a solution containing 1% polyvinyl alcohol (Sigma-Aldrich, Burlington, MA, USA) and, subsequently, by lysis with pre-warmed Aqua Braun (B. Braun, Melsungen, Germany). The resulting neutrophils were cultured in DMEM medium supplemented as above. The purity of the neutrophil population after isolation was routinely >98%.

### 4.11. Apoptosis Assays

Neutrophils (10^6^ cells/mL) were stimulated as indicated and were stained 24 h later with FITC Annexin V/propidium iodide according to the manufacturer’s instructions (BioLegend). Quantification was performed with a BD FACSCanto II flow cytometer (BD Biosciences, Heidelberg, Germany).

### 4.12. Statistical Analysis

Clinical data were analyzed with the SPSS statistical software version 26 (IBM Corporation). Survival curves (5-year, 3-year or 1-year cut-off) were plotted according to the Kaplan–Meier method. Significance was initially tested by univariate analysis using the log-rank test. Multivariate analysis was subsequently used to determine the prognostic value of selected variables using Cox’s proportional hazard linear regression models adjusted for age, Karnofsky Performance Scale (KPS), therapy, extent of surgical resection and MGMT methylation status. The in vitro data were analyzed with the paired student’s *t*-test. In all studies, the level of significance was set at *p* ≤ 0.05.

## Figures and Tables

**Figure 1 ijms-23-06037-f001:**
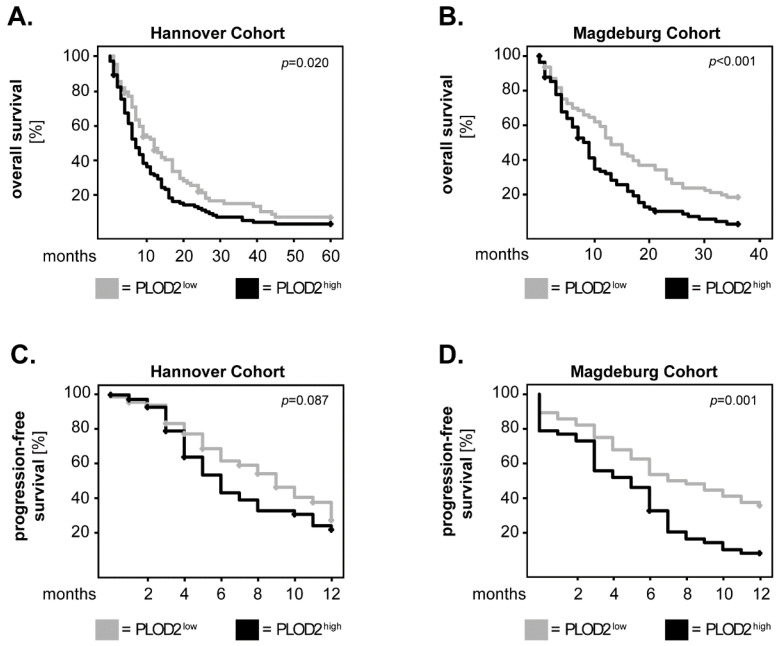
**PLOD2 in GBM patients: univariate analysis of survival.** PLOD2 expression was dichotomized into PLOD2^high^ and PLOD2^low^ according to the median-split method. Kaplan–Meier curves were plotted for the (**A**) 5-year overall survival (OS) in the Hannover cohort, (**B**) 3-year OS in the Magdeburg cohort and (**C**,**D**) 1-year progression-free survival (PFS) in both cohorts. The log-rank test was used for statistical analysis and the *p*-values are indicated in the upper-right corner of each plot.

**Figure 2 ijms-23-06037-f002:**
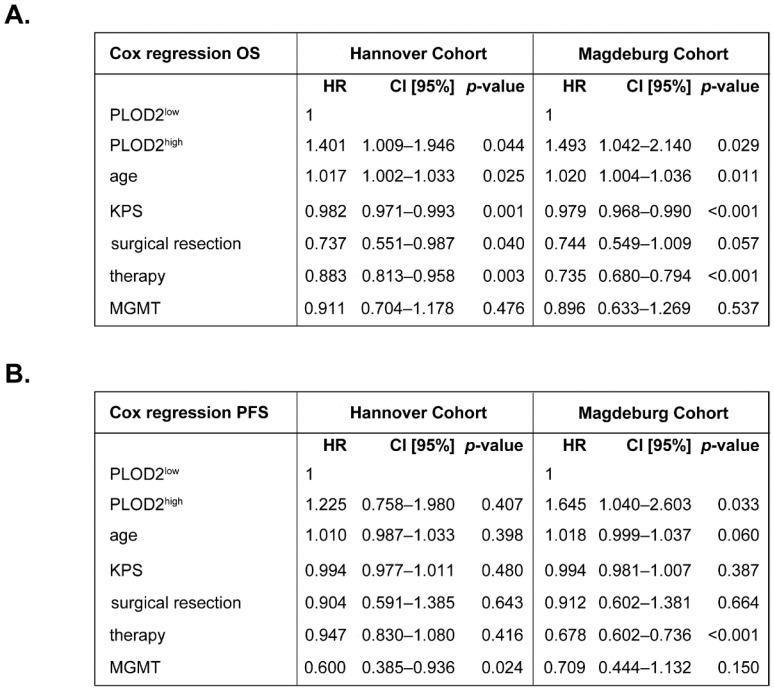
**PLOD2 in GBM patients: multivariate analysis of survival.** Multivariate Cox regression analysis model for the (**A**) overall survival (OS) and (**B**) progression-free survival (PFS) of patients with high versus low levels of PLOD2. HR: hazard ratio; CI [95%]: 95% confidence interval.

**Figure 3 ijms-23-06037-f003:**
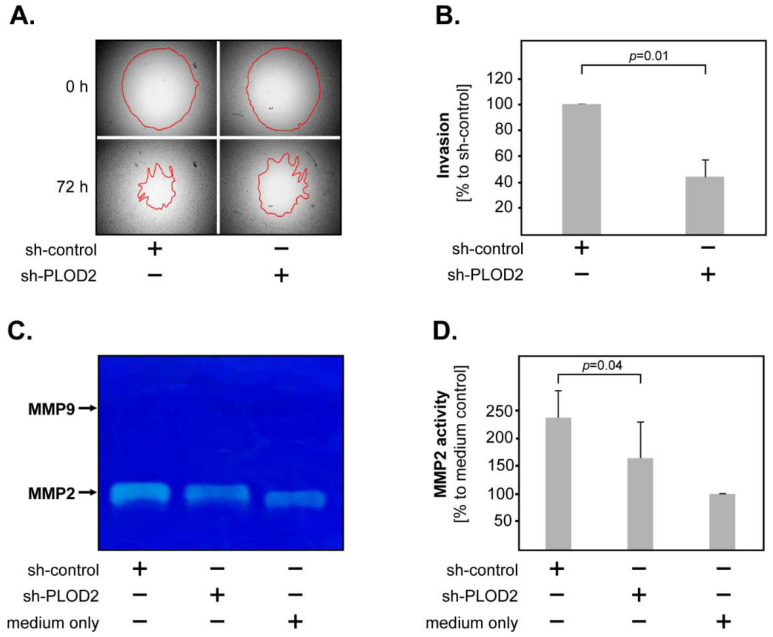
**PLOD2 and GBM invasion.** (**A**) Representative micrographs of invasion assays in sh-control versus sh-PLOD2 H4 cells. The upper panels show the pre-invasion status at 0 h, the lower panels show the post-invasion status at 72 h. The red line marks the closure of the “gap”, indicating the degree of tumor invasion. (**B**) PLOD2 knock-down significantly reduces the invasiveness of H4 cells. The data are presented as percentage to sh-control. (**C**) Representative image of a gelatin zymography gel showing that both sh-PLOD2 and sh-control released MMP2, but only negligible levels of MMP9. (**D**) PLOD2 knock-down significantly reduces the release of MMP2 in H4 cells. For quantification, the integrated density of the bands was determined using the ImageJ software. The data are presented as percentage to medium only. Shown are the means + S.D. of 3 independent experiments. Statistical analysis was performed with the paired *t*-test.

**Figure 4 ijms-23-06037-f004:**
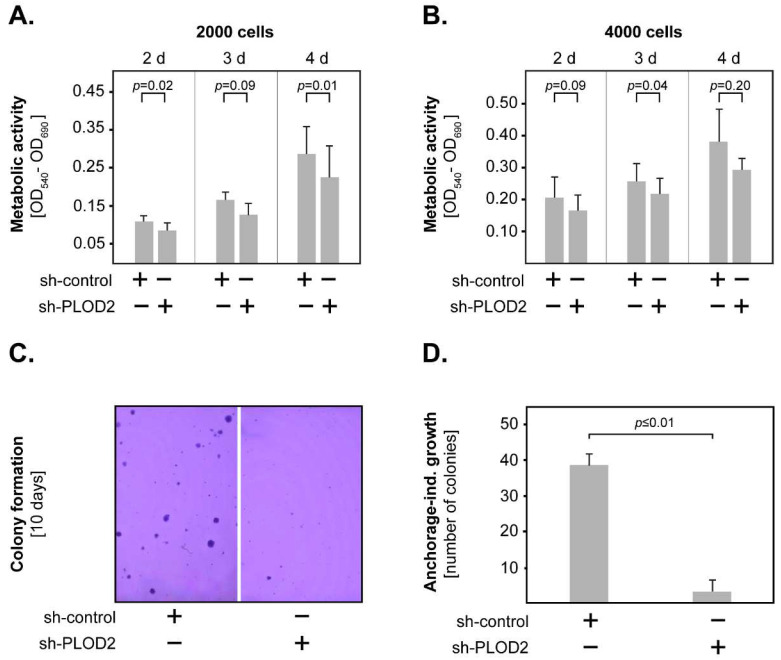
**PLOD2 and GBM proliferation.** PLOD2 knock-down reduced the metabolic activity of H4 GBM cells, as indicated by the MTT assay using (**A**) 2000 cells and (**B**) 4000 cells. (**C**) Representative micrographs of colonies generated by sh-control and sh-PLOD2 cells after 10 days of culture in low-gelling agarose. (**D**) PLOD2 knock-down significantly inhibited the anchorage-independent growth of H4 cells. Shown are the means + S.D. of at least three independent experiments per assay. In all studies, statistical analysis was performed with the paired *t*-test.

**Figure 5 ijms-23-06037-f005:**
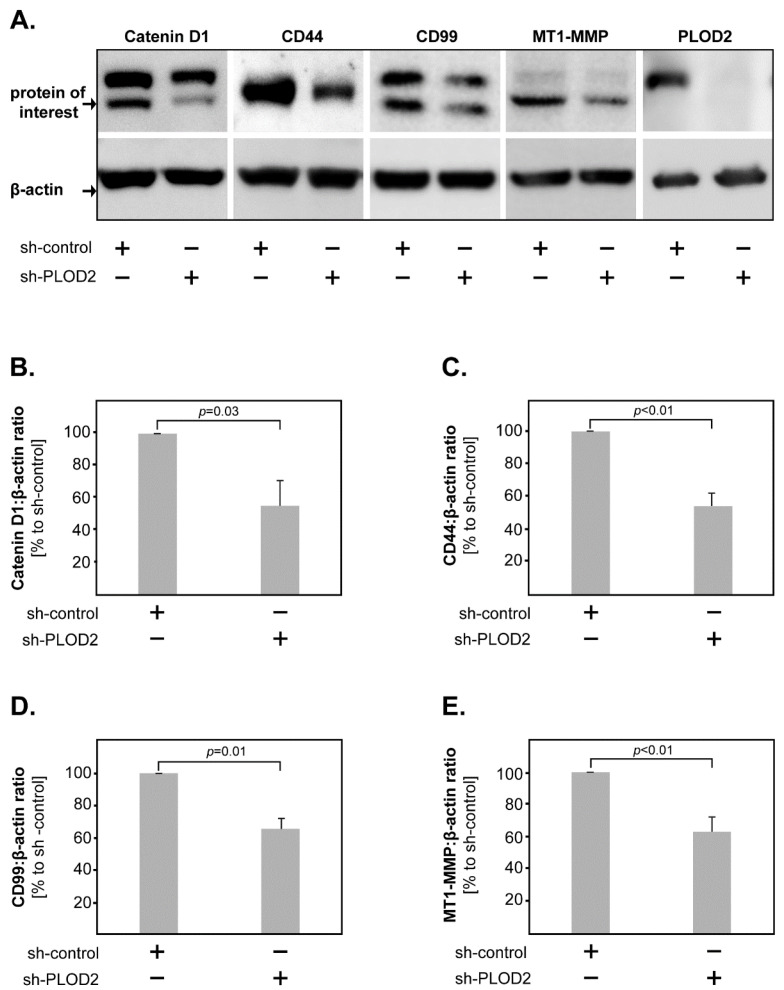
**Molecular mechanisms of PLOD2 in GBM.** (**A**) Representative western blots of Catenin D1, CD44, CD99, MT1-MMP and PLOD2 levels in sh-PLOD2 versus sh-control GBM cells. Beta-actin was used as a loading control. PLOD2 knock-down significantly decreased the levels of (**B**) Catenin D1, (**C**) CD44, (**D**) CD99 and (**E**) MT1-MMP in GBM cells. Shown are the means + S.D. of at least three independent experiments. Statistical analysis was performed with the paired *t*-test.

**Figure 6 ijms-23-06037-f006:**
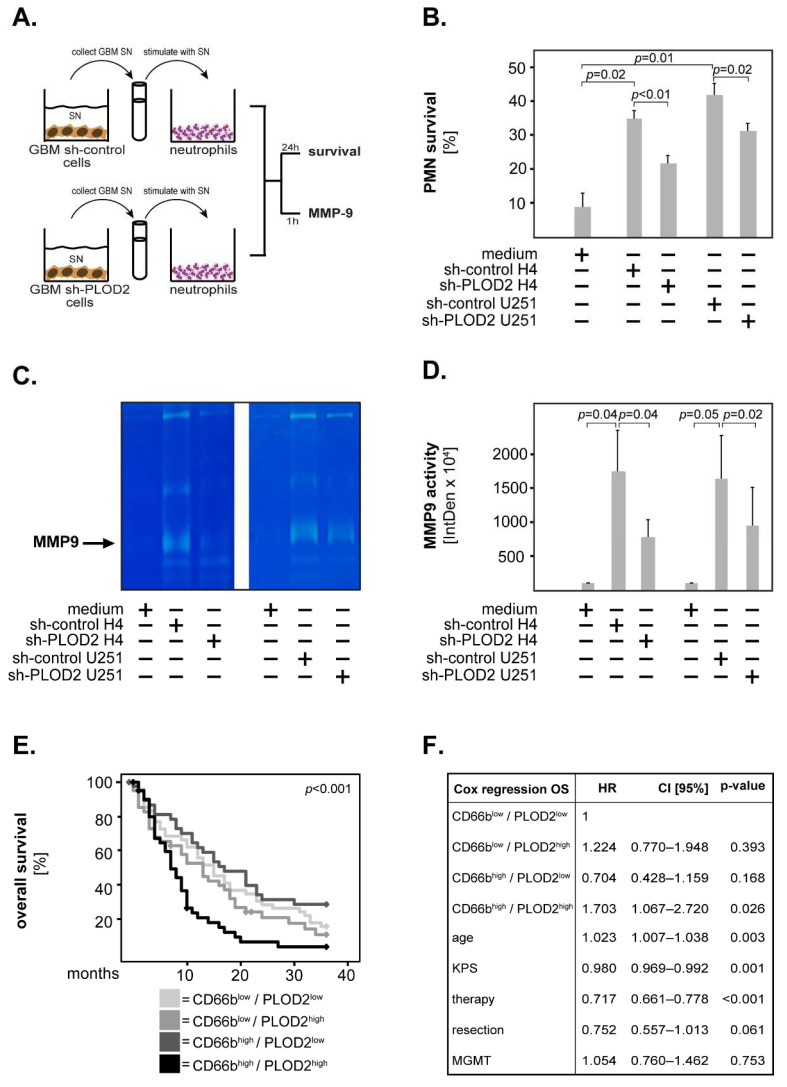
**PLOD2 and neutrophils in GBM.** (**A**) Neutrophils were incubated with supernatants (SN) derived from sh-control or sh-PLOD2 GBM cells. Culture medium was used as control. The survival of neutrophils was assessed at 24 h and the release of MMP9 at 1 h post-stimulation. (**B**) Sh-control SN prolonged the survival of neutrophils in both cell lines. This effect was significantly lower upon stimulation with PLOD2 knock-down SN. (**C**) Representative image of a gelatin zymography gel showing that neutrophils release MMP9 upon stimulation with GBM SN. (**D**) Neutrophils stimulated with sh-PLOD2 SN released significantly lower levels of MMP9 compared with their sh-control stimulated counterparts. For quantification, the integrated density of the bands was determined using the ImageJ software. The data are presented as percentage to medium only. Shown are the means + S.D. of three independent experiments. Statistical analysis was performed with the paired *t*-test. (**E**) GBM patients were divided into four groups according to the combined expression of CD66b and PLOD2. The Kaplan–Meier curves were plotted for the 3-year overall survival and the statistical significance was determined with the log-rank test. The *p*-value is indicated in the upper right corner of the plot. (**F**) Multivariate Cox regression analysis model for the overall survival of the four groups of patients. HR: hazard ratio; CI [95%]: 95% confidence interval.

**Figure 7 ijms-23-06037-f007:**
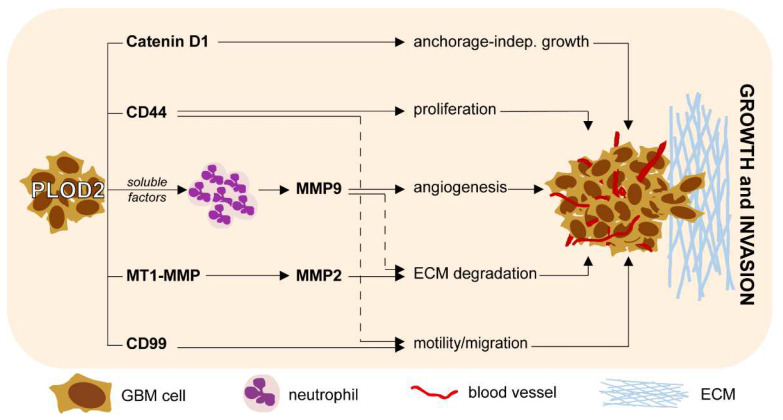
**Schematic representation of the main findings integrated in the biology of GBM**.

## Data Availability

Data from the in vitro studies are contained within the article and Appendix A. The datasets involving GBM patients are available from the corresponding author on reasonable request.

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
