# Peer review of "PLOD2 Is a Prognostic Marker in Glioblastoma That Modulates the Immune Microenvironment and Tumor Progression"

_ijms, 2022, doi:10.3390/ijms23116037_

Round 1
Reviewer 1 Report
General comments to the Authors
Despite the positive reviews of the original versions of the manuscript, there are glaring weaknesses that significantly diminish enthusiasm for its potential clinical utility. First, the study lacks requisite statistical power and replication to reliably validate the accuracy and reproducibility of its results and conclusions. Second, the study lacks requisite and validating control groups utilizing established PLOD2 and CD66b inhibitors to establish relative selectivity of action. Third, the study is largely confirmatory of a previously published study by Oncotarget. 2017 Apr 4;8(14):23401-23413.; Oncotarget. 2021 Jul 6;12(14):1442-1443.; Biomed Pharmacother. 2017 Jun;90:670-676. and therefore lacks significant novelty.
Author Response
Please see attached WORD file: Point-by-Point Reply REVIEWER 1

Reviewer 2 Report
This study provide first evidence that PLOD2 can modulate the immune microenvironment in GBM by inducing neutrophil granulocytes to acquire a pro-tumor phenotype. Thus, PLOD2 is a tumor-promoting factor in GBM which may serve as a potential target for novel therapeutic strategies in this type of cancer.
I have some questions, suggestions which may help improve the manuscript.
- The authors write that „During tumor progression, increased collagen crosslinking promotes stiffening of the extracellular matrix, thus enhancing invasion and metastasis.” In a later subsection they mention that „PLOD2 promotes release of extracellular matrix-degrading MMP2 – a mechanism associated with enhanced invasiveness” They are opposite mechanisms. Both of them are important in enhanced invesivness?
- The Figure 2. looks like as a Table.
- The sentence is not entirely clear: lane 115-117 „To this end, the cells were stably transfected to downregulate the levels of PLOD2 (sh-PLOD2) or with a control plasmid (sh-control)”. The cells were stably transfected with …. to downregulate the leveles of PLOD2.
- What is the difference between the H4 GBM cell line and the former mentioned U87 and U251 GBM cells?
- In case of Figure 3.-8. please explain briefly the meaning of notations: like sh-control + - - and their complex combination.
- The authors mention that: lane 124 „We found that H4 GBM cells released MMP2 but only negligible levels of MMP9” In a later subsection they mention: lane 180-183 „GBM cells modulate the biology and functions of neutrophils via PLOD2. To test this hypothesis, we produced conditioned supernatants (SN) from sh-control and sh-PLOD2 GBM cells. Subsequently, we stimulated peripheral blood neutrophils with these supernatants and determined neutrophil survival as well as the release of MMP9 – both indicators of a pro-tumor neutrophil phenotype” Why did you analyse the release of MMP9 instead of MMP2?
Author Response
Please see attached WORD file: Point-by-Point Reply REVIEWER 2

Round 2
Reviewer 1 Report
Majority review report for a manuscript entitled “PLOD2 is a prognostic marker in glioblastoma that modulates the immune microenvironment and tumor progression” submitted by the first author Nina Kreße and corresponding authors Claudia Alexandra Dumitru. The authors ordered less positive data fitting their advocation. However, this study is too immature, and many data are scientifically questionable.
- This original study is largely confirmatory of a previously published study by 2017 Apr 4;8(14):23401-23413.; Oncotarget. 2018 Jan 25;9(10):9400-9414.; Oncotarget. 2017 Jun 27;8(26):41947-41962.; Biomed Pharmacother. 2017 Jun;90:670-676.; PLoS One. 2021 Jan 27;16(1):e0246097. and therefore, lacks significant novelty of this original study.
- The author should explain the significance of procollagen-lysine 2-oxoglutarate 5-dioxygenase 2 (PLOD2) as a promising therapeutic target for glioma migration and invasion in the result section more in the discussion section especially their clinical significance. We also would like to point out that the findings in this paper cannot be explained mechanistically.
- The "Discussion" section needs to be re-written. It is important to "discuss" various issues but not to “re-describe" the findings in this section.
- Please correct all of the grammatical mistakes present in the manuscript.
Author Response
Please see attached WORD file: 2. Revision_Point-by-Point Reply to REVIEWER 1
